# The Implementation of Interactive VR Application and Caching Strategy Design on Mobile Edge Computing (MEC)

**Shu-Min Chuang \*** , **Chia-Sheng Chen and Eric Hsiao-Kuang Wu**

Department of Computer Science and Information Engineering, National Central University, Taoyuan City 320317, Taiwan; 108522122@cc.ncu.edu.tw (C.-S.C.); hsiao@csie.ncu.edu.tw (E.H.-K.W.)
\* Correspondence: shumin.chuang@ttc.org.tw

**Abstract:** Virtual reality (VR) and augmented reality (AR) have been proposed as revolutionary applications for the next generation, especially in education. Many VR applications have been designed to promote learning via virtual environments and 360° video. However, due to the strict requirements of end-to-end latency and network bandwidth, numerous VR applications using 360° video streaming may not achieve a high-quality experience. To address this issue, we propose relying on tile-based 360° video streaming and the caching capacity in Mobile Edge Computing (MEC) to predict the field of view (FoV) in the head-mounted device, then deliver the required tiles. Prefetching tiles in MEC can save the bandwidth of the backend link and support multiple users. Smart caching decisions may reduce the memory at the edge and compensate for the FoV prediction error. For instance, caching whole tiles at each small cell has a higher storage cost compared to caching one small cell that covers multiple users. In this paper, we define a tile selection, caching, and FoV coverage model as the Tile Selection and Caching Problem and propose a heuristic algorithm to solve it. Using a dataset of real users' head movements, we compare our algorithm to the Least Recently Used (LRU) and Least Frequently Used (LFU) caching policies. The results show that our proposed approach improves FoV coverage by 30% and reduces caching costs by 25% compared to LFU and LRU.

**Keywords:** edge caching; 360 live streaming; 5G standalone network; multiple tiles; virtual reality

## 1. Introduction

Virtual reality (VR) and augmented reality (AR) have significant potential as future applications. A review of virtual reality in education [1] has shown that an increasing number of VR applications are being designed for rigorous training or realistic simulations. For instance, VR applications in fire safety [2] and pyramid tours are more immersive for users than teaching videos or textbooks. Additionally, 360 video brings more application scenarios to VR. We have designed a mission-aware VR question application that can customize the question according to the virtual environment. However, the disadvantage of 360 video streaming and VR is that they are limited by various environmental factors, which reduces their training effectiveness.

In the past, experiencing virtual environments involved a high cost for the required devices, such as head-mounted displays (HMDs), high-end computers, etc. Moreover, VR headsets require a connection to a computer with image and audio transportation, which restricts users within a certain range of movement. Recently, more and more mobile HMDs and cardboard constructs that combine VR headsets with smartphones have been designed, such as the HTC Focus, Oculus Quest, and Google Cardboard [3]. These devices involve little computation capacity to decode video packets, then render them on-screen. Such devices can replace expensive high-end equipment, bringing a new type of VR service. However, delivering high-quality content will be a challenge to the network in the future.

According to Cisco's visual networking index report [4], AR and VR traffic was expected to reach 254 petabytes per month in 2022. In more detail, VR streaming requires

290 Mbit/s transfer for streaming 12 K 360° video at 60 frames per second (FPS) for an ideal experience. To ensure the quality of the user experience, HMD frame rates of up to 90 FPS for Cloud VR gaming and 30 FPS for VR video are recommended [3]. It is hard to meet such strict requirements in the current network.

Streaming full 360° video involves an unessential cost in the network. When general HMDs receive 360 frames, they project onto a spherical surface, then display the field of view (FoV) on the headset's screen; this only takes about 53% of the required bandwidth for 360° video streaming. One of the streaming strategies to deliver the user's field of view (FoV) is called tile-based live streaming as shown in Figure 1. This strategy cuts a 360° image into certain-size segments called tiles. The streaming service receives the user's view, then transmits the relevant tiles to users. After receiving the whole tiles, the HMD combines the tiles and displays them on the screen.

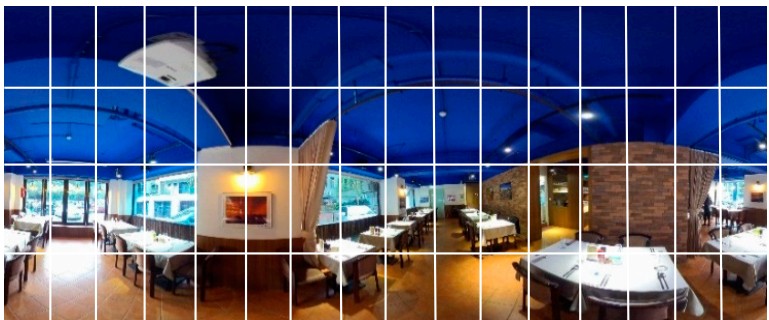

**Figure 1.** A tile-based image.

A new problem that arises in tile-based streaming is that the total response time, including the headset motion updating time, video encoding time, delivery time, etc., needs to be finished within the HMD's updating frame rate. If the responding time is larger than the HMD's updating frame rate, the service cannot transmit the frame according to the current viewpoint immediately. It may deliver the wrong FoV tiles to the client, then display nothing at the edge of the screen, which we call black edge. The presence of a black edge may decrease the quality of the experience for the client.

Recently, Mobile Edge Computing (MEC) was proposed for providing computing and caching resources close to the client. For latency-sensitive applications, such as autonomous vehicles and VR streaming, the service provider can deploy their service or cache their content at the edge to shorten the latency. For tile-based streaming, caching the relative content files at the edge not only shortens the latency between services and users but also reduces the bandwidth of the backhaul link by providing the same content to multiple users.

With the tiling technique and the storage capacity, the required tiles can be pre-fetched in MEC. Deciding the tile size and selecting which tiles are pre-cached in which data center can be modeled as a minimum resource and great Quality of Experience (QoE) problem. The larger the size of the tile, the higher the probability of being shared. For example, we can cache all required tiles in a small cell. However, the coverage of the small cell may duplicate coverage for the same user, which expends extra caching costs. We model this challenge as the Tile-based Selection and Caching Problem, and we propose a heuristic algorithm to solve it.

## 2. Related Work

VR applications have been designed for multiple scenarios in recent years. Yap, M.C. [5] designed an immersive VR module that provides 360° video within a learning management system (LMS). C. Malinchi et al. [6] implemented an immersive system with panoramic-based imagery and provided virtual library resources (books) and their metadata. K. Zhang et al. [2] built a 3D virtual environment for a fire safety education system based on VR technology. H. Kim et al. [7] combined Virtual Reality and an education system that provides VR content to students, in which their learning performance can



be monitored by a supervisor. Ahmed et al. [8] designed a virtual zoo that delivers 3D computer-generated animal objects with virtual surroundings using animations and videos from the real environment. H.-H. Tsai et al. [9] built an interactive VR system of soil and water conservation in outdoor classrooms.

Edge caching is an important technique for supporting both video streaming and video on demand. To solve the black edge problem in FoV streaming, many studies have proposed farsighted tile selection and caching strategies. For tile selection, C. Ozcinar et al. [10] proposed tile-based streaming combined with the DASH standard. With its adaptive bitrate, the system was able to deliver high-resolution 360° video and maintain the quality of the experience. Alireza Zare et al. [11] proposed storing two versions of the same video with high resolution and low resolution. The service transmits the user's FoV in high resolution and covers the area outside of the FoV with the low-resolution version. C.H. Hsu [12] provided multi-resolution images according to the distance from the user FoV to cover the area outside of that shown on the screen. Regarding caching policy, J. Shi et al. [13] proposed a greedy algorithm to solve the joint tile-based caching, transmitting, and transcoding problem. J. Zhang et al. [14] proposed a replacement strategy with Scalable Video Coding (SVC) for when the caching capacity reaches its limit. P. Maniotis and N. Thomos [15] adopted a Long Short-Term Memory network for predicting the popularity of the tiles, then caching at the edges. A. Mahzari et al. [16] proposed an FoV-aware caching policy based on the users' FoVs. The FoV-aware caching policy learns a probabilistic model of common FoVs for each 360° video based on previous users' viewing histories to improve caching performance. S. Kumar et al. [17,18] proposed dynamic partitioning and popularity-based caching for optimized performance in content-centric fog networks. The scheme partitions the fog network by grouping the fog nodes into non-overlapping partitions to improve content distribution in the network and to ensure efficient content placement decisions. Moreover, G. Gaurav et al. [19] proposed resource scheduling in the fog environment using optimization algorithms for 6G networks. This work makes a comparison of PSO, GA, and Round-Robin algorithms on parameters, such as cost, makespan, average execution time, and energy consumption among the group behavior species, social behavior, and preemptive type, which is better for achieving QoS for resource management in the fog environment for the 6G network.

## 3. Interactive VR Application

In this section, we introduce a mission-aware VR application called VR Quick Question, a modular system in which a teacher can customize questions, then produce pin codes and QR codes for students. The deployment of panoramic images showing scenes from around the world makes students more interested in learning geography-related knowledge.

### 3.1. System Design

VR Quick Question is a modular system consisting of two parts: the client side and the supervisor side. On the supervisor side, teachers can design question scenarios from a few prepared templates and monitor the answering status of students viewing the questions they designed. On the client side, students can choose to enter the website or install the application on their smartphone to start the questions. The application provides two play modes: tablet mode and cardboard mode. In tablet mode, students can play by touching and pressing as in the normal use of a smartphone. In cardboard mode, a smartphone-based VR headset is supported using Google Cardboard. The student can answer questions by using the white spot to select options. This cardboard setup is lightweight and less expensive compared to a regular VR headset.

### 3.2. Customized Question Scenarios

We prepared three types of question scenario templates: text, panorama, and 360 video scenarios, shown in Figure 2. Each template can be used to deploy multiple types of questions. The text scenario shows a window displaying multiple types of questions, such

as multiple-choice questions, sorting questions, and questions with pictures. Each question type can be customized according to the subject of the question. The panorama scenario displays a panoramic image surrounding the user. It can deploy question points on the panoramic image at specific positions. The 360° video scenario displays a 360° video surrounding the user. The teacher can set questions at specific timespans that may be related to the scene from a few seconds prior.

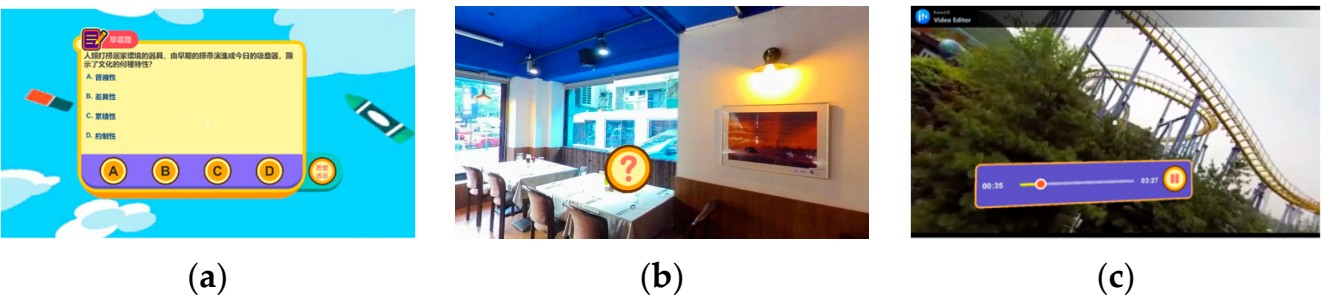

| **(a)** | **(b)** | **(c)** |

**Figure 2.** (**a**) Text Scenario; (**b**) Panorama Scenario. The question mark in the middle represents the question point deployed on the image; (**c**) 360° video scenario.

### 3.3. System Architecture

The whole system is comprised of a user module, a monitoring module, and a question customizing module. The user module is responsible for collecting basic user information and tracking their answer status. The monitoring module allows the teacher to supervise the performance of the students, including their viewpoints and chosen answers. The question customizing module enables teachers to create unique questions. On the server side, teachers can deploy self-made panoramic images for the questions and set question points based on specific landmarks or buildings. For instance, a panoramic picture of Taipei 101 can be projected, and a question point can be set on the building. The designed question, in this case, might be to choose the correct name of the building. After setting the question, the data is packed as a JSON file, and a unique pin code and QR code are generated. The student then enters VR Quick Question and inputs the pin code or scans the QR code. The client-side loads the relevant JSON file, renders the panoramic image, and displays the question point on the screen.

## 4. Model and Problem Description

In this paper, we focus on the tile selection and caching strategy. The size of the tiles may cause a large difference in the caching policy. There is a tradeoff between the caching cost and FoV coverage. Caching all tiles of one 360° image will cover all possible angles of FoV, but it will cause unnecessary storage if the tiles are not required. First, we describe tile-based streaming architecture in greater detail and model the problem of the tile selection and caching strategy.

### 4.1. Tile-Based Streaming Architecture

We consider a novel tile-based video streaming architecture, shown in Figure 3 and the notations were described in Table 1. When users start live broadcasting, 360° panoramic images are captured by omnidirectional cameras, then transmitted continuously to the streaming service. Each equirectangular image is cut into tiles of a certain size. We define the yaw and the pitch of the tile as $N_y$ and $N_p$, respectively. According to the side length of the tile, the total tiles of one equirectangular image can be defined as $L = 360/N_y * 180/N_p$, in which the yaw and the pitch of the equirectangular image are equal to 360° and 180°.

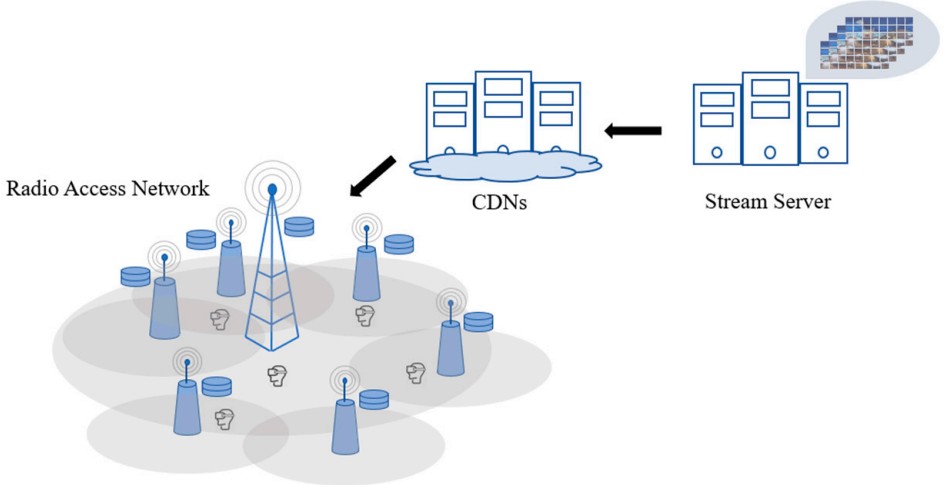

**Figure 3.** Tile-based streaming architecture.

**Table 1.** NOTATION.

| Symbol | Definition |
|---|---|
| $t$ | Time duration of one GOP |
| $V$ | Number of data centers (DC) |
| $K$ | Number of users |
| $T_p$ | Number of pre-cached GOPs |
| $N$ | Size of tile |
| $N_y$ | Degree of yaw of a tile |
| $N_p$ | Degree of the pitch of a tile |
| $L$ | Number of tiles |
| $X_i^{l,t}$ | Binary value. Caching decision at DC $i$ with tile $l$ for time $t$ |
| $Y_j^{l,t}$ | Binary value. Tile $l$ is cached at a set of base stations that cover user $j$ |
| $F_j^{l,t}$ | Binary value. Required tile for user $j$ at time $t$ |

The size of one tile can be defined as $N = N_y * N_p$. After transcoding the tiles, all tiles are cached in Content Delivery Networks (CDNs). The future FoV of users is predicted, and then the required tiles are selected for users. The required tiles are precached in data centers in MEC by the heuristic algorithm, which will be presented in a later section. Each frame is combined as segments with a fixed time duration. For instance, to transmit a 60 fps 360° video, 60 frames are gathered in a segment with a 1-s time duration. When users start watching a video, the required tiles which cover the FoV are requested by the users. If the required tiles are not cached at the base station which connects with the user, the tiles will be delivered from a CDN, taking a long time to arrive at the user. On the other hand, we can cache all tiles at the base station. With this decision, many redundant tiles that are not required will cause unnecessary storage. For the next section, we model this problem as the Caching and FoV Coverage Model.

### 4.2. Caching and FoV Coverage Model

In this section, we focus on the caching strategy in MEC, shown in Figure 3 for a Radio Access Network (RAN). We model the caching cost, communication cost, and FoV coverage as our objective function. In modeling the caching cost, we denote $X_i^{l,t} \in \{0, 1\}$

as the caching decision in MEC. $X_i^{l,t} = 1$ when tile $l$ is pre-cached in datacenter $i$ at time duration $t$, and $X_i^{l,t} = 0$ otherwise. Therefore, the caching cost can be represented as:

$$\delta_S = \sum_{i=0}^{V} \sum_{l=0}^{L} X_i^{l,t} * N \tag{1}$$

As the caching constraint, each data center has the maximum caching capacity $\mu_s^{max}$. The total size of cached tiles cannot exceed $\mu_s^{max}$. We can formulate this constraint as:

$$\sum_{r=t}^{t+T_p} \sum_{l=0}^{L} X_i^{l,r} * N \le \mu_s^{max}, \ \forall i,t \tag{2}$$

where $T_p$ represents the time duration of each pre-cached tile in the data center. For example, $T_p = 1$ means that the datacenter has pre-cached one Group of Picture (GOP) of tiles for the future FoV.

For FoV coverage, we define it as a positive value, meaning that more coverage receives a greater score. The equation can be shown as:

$$\delta_F = \sum_{j=0}^{K} \sum_{l=0}^{L} \frac{Y_j^{l,t} * F_j^{l,t}}{F_j^{l,t}} \tag{3}$$

where $F_j^{l,t}$ denotes the total required tiles of the FoV for user $j$. $F_j^{l,t} = 1$ when tile $l$ is required by user $j$, and $F_j^{l,t} = 0$ otherwise. $Y_j^{l,t}$ represents a binary value. If tile $l$ is cached at a set of base stations that cover user $j$, then $Y_j^{l,t} = 1$.

Eventually, the problem of the tile selection and caching strategy can be formulated as:

$$\min \alpha \delta_S - \gamma \delta_F \\ s.t. \ (1) \tag{4}$$

## 5. Proposed Methods

### 5.1. Tile Selection

By deciding the length of $N_y$ and $N_p$, we first define the rule of tile selection. To cover the FoV, the total size of selected tiles must be larger than the screen size of the HMD. In this rule, the ranges of the extra size of yaw and pitch should be defined at $0°$ to $N_y$ and $0°$ to $N_p$, respectively, in which $N_y$ and $N_p$ are smaller than the HMD screen. After that, we calculate the average and standard deviation of the degree of yaw and pitch of the head movement, then transform them into $N_y$ and $N_p$. We formulate the equations as:

$$N_y = \frac{\overline{x}_y + \sigma_y}{1000} * fps \tag{5}$$

$$N_p = \frac{\overline{x}_p + \sigma_p}{1000} * fps \tag{6}$$

Next, we introduce Time-aware Tile Selection. According to the analysis result of tiles and the previous viewpoint, we estimate the future viewpoint and the required tiles for the Group of Pictures. To simplify the problem, we ignore the prediction of the viewpoint. For the required tiles, we modify the selection algorithm according to the time duration of GOP $T_g$. The larger the time duration of the GOP, the greater the range of the FoV that should be covered. We formulate the equations as:

$$M_y = \frac{\overline{x}_y * T_g}{1000 * N_y} \tag{7}$$

$$M_p = \frac{\overline{x}_p * T_g}{1000 * N_p} \qquad (8)$$

*5.2. Overlap-Aware (OA) Algorithm*

We design a heuristic algorithm according to the requirement of each tile. From the current user viewpoint, the streaming service can predict the required tiles for the next segment. By counting the requirement of each tile, we can set its priority. If the total requirement of the tile exceeds half of the sum of the maximum and minimum tile requirements $M_l^{max}$ and $M_l^{min}$, we set it to high priority. Otherwise, if the requirement is lower, we set the tile to low priority. User $j$ is surrounded by a set of base stations, $Vj$. For each tile $l$ that is required by user $j$, we cache tile l at the maximum-connection base station in $Vj$ if the tile $l$ is a high priority. If the maximum-connection base station exceeds its storage capacity, the tile will be cached at the second-most-connected base station, and so on. If the tile is a low priority, we choose the base station with the maximum available capacity. Algorithm 1 shows the overall Overlap-aware algorithm in pseudocode.

---

**Algorithm 1** Caching decision

---

**for** user $j \in K$

Time duration of one GOP

 **for** tile $l \in$ total required tiles $L_j$ for user $j$

  $M_l \leftarrow$ Calculate the number of requests of tile $l$

  **if** tile $l$ is already cached at $V_j$

   **continue**

  **end**

  **if** $M_l < \left( M_l^{min} + M_l^{max} \right)/2$

   $i \leftarrow$ Pick the base station with the maximum available

    capacity $\in V_j$

   $X_i^l = 1$

  **else**

   $i \leftarrow$ Pick the base station with the strongest connection and

    sufficient capacity $\in V_j$

   $X_i^l = 1$

  **end**

 **end**

**end**

---

## 6. Experimental Setup

The experiment was conducted in two phases. In the first phase, we chose the dataset provided by [20] and built a simulation environment in MATLAB. This dataset holds rotation data of 60 users' head movements and viewpoints within 60 s. Moreover, the viewpoints in this dataset were recorded in multiple types of videos. The videos in this dataset are classified into fifteen categories corresponding to the camera movement and several moving targets. We selected the head movement videos corresponding to a fixed camera and single moving targets.

We set the FPS of the 360° videos at 60 fps and the time duration of one GOP equal to 1 s. The screen size was set at 110° and 90° for the degrees of yaw and pitch, respectively. With the parameters above, we set the yaw and pitch of tiles and the extra margin out of the FoV. The yaw and pitch of tiles were set at 30° and 10°, respectively. The extra margin of tiles was set at 2 tiles for yaw and pitch.

For the network topology, we set 480 users and 19 base stations. If the distance between a user and a base station was shorter than the cover range of the base station, the user was connected to the base station. The cache capacity was 50% of the total size of the video.

For the caching algorithm, we compared our proposed algorithm with the Least Frequently Used (LFU) and Least Recently Used (LRU) caching algorithms. In LFU caching

policy, the operator evicts tiles with lower access frequency when a base station exceeds its maximum storage capacity. In LRU caching policy, the operator evicts the tiles with the longest time duration since the last access.

In the second phase, we set up mobile communication for the interactive VR application based on a 5G Standalone (SA) Network, using the Nokia Core Network and Mobile Edge Computing for this trial testbed as illustrated in Figure 4. The 5G Core Network functions include the Access and Mobility Management Function (AMF), Authentication Server Function (AUSF), Session Management Function (SMF), Unified Data Management (UDM), and User Plane Function (UPF). For the Mobile Edge Computing network function, we implemented the User Plane Function (UPF) to handle the local equipment data of the interactive VR application and the delivery of all data to the application server for 360 live streaming and edge content caching services. The transmission of the 5G Trial network was in the 3.7~3.8 GHz band. The interactive VR application was run on a high-performance computing stream server (High-Performance Computing Server, HPC), in which complex calculations could be performed immediately to make the terminal device present a smooth image, and the specific development platform software was used to transmit the VR (Virtual Reality) or MR (Mixed Reality) screen. In the streaming system architecture, we used CentOS 7 as the underlying system, and the middle layer was established through the virtual operating platform OpenStack Train to set up a virtual Windows 10 system. The hardware system included an NVIDIA RTX A4000 graphics processing unit (GPU), through NVIDIA GPU VR-Ready. Terminal VR devices can generate a three-dimensional virtual world to achieve virtual–real interaction based on a Qualcomm Snapdragon XR2 chip and support the 5G communication mode.

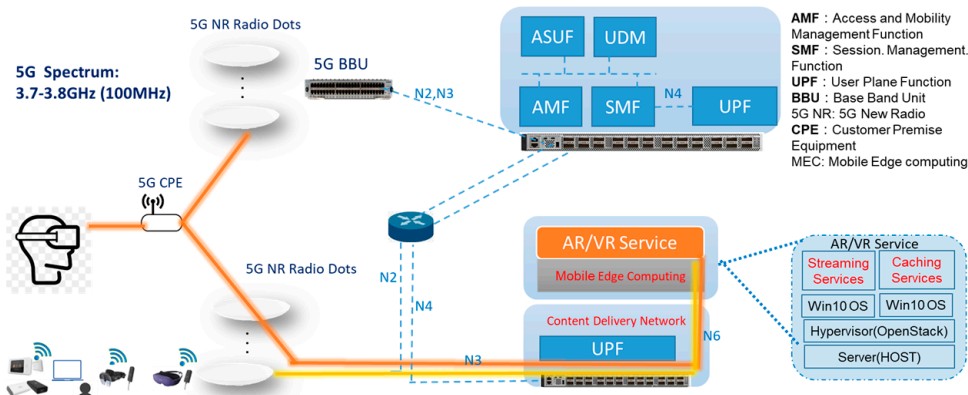

**Figure 4.** 5G Standalone (SA) Network and Mobile Edge Computing.

According to K. Salah et al. [21], more sophisticated caching algorithms are necessary for mobile AR/VR in 5G networks. One such approach is content-aware caching, which takes into account the specific characteristics of AR/VR data. This type of algorithm considers factors such as the spatial and temporal locality of content and makes caching decisions accordingly. By understanding the patterns and properties of AR/VR data, content-aware caching algorithms can optimize caching strategies for improved performance.

In the context of developing caching techniques in 5G cellular networks, C. Mehteroglu et al. [22] identify three main issues that need to be addressed: what to cache, where to cache it, and when to release it. In 5G networks, there are two fundamental methods for cache deployment. The first method involves placing caches within the 5G Core (5GC) network. By deploying caches within the 5GC, where they serve multiple users and handle a significant portion of the traffic, the cache's success rate can be improved. Deploying caches within the 5GC network offers advantages in terms of deployment and maintenance ease. Alternatively, caches can be deployed at the radio access network (RAN), which consists of gNodeBs (5G base stations). Caches are connected to these gNodeBs. To avoid redundancy, neighboring gNodeBs should not store the same content, as they can share information with end consumers. T. Ismail et al. [23] and J.A. Kakar [24] suggest

that proactive caching near users in the RAN has been a successful method for reducing backhaul traffic load and latency in 5G networks.

S. Anokye et al. [25] propose that mobile edge networks can address the increasing demand for bandwidth, congestion in backhaul networks, higher operating and maintenance costs, and performance issues. Therefore, we propose relying on tile-based 360° video streaming and the caching capacity in Mobile Edge Computing (MEC) to predict the field of view (FoV) in the head-mounted device and deliver the required tiles for the Interactive VR Application.as illustrated in Figure 5.

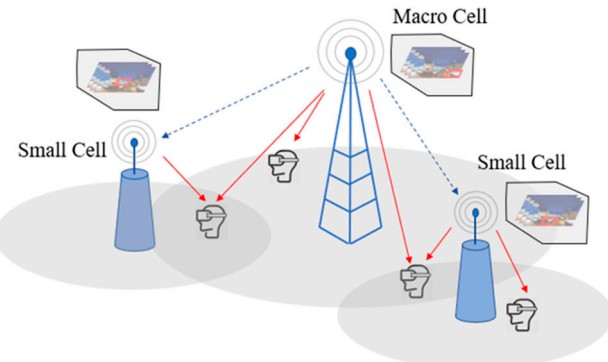

**Figure 5.** Caching Scenario at Mobile Edge Computing.

## 7. Performance Evaluation

### 7.1. Hit Ratio

The hit ratio is one of the indicators used to evaluate a caching policy. This value can be represented by the percentage of cache hits out of cache hits plus cache misses. A cache hit means that the tile required by the user was kept in the cache. If the required tile was not kept in the cache, it is called a cache miss. In a one-on-one connection, when the base station is missing a tile, extra time is required to transfer the required tile from a CDN. The results in Figure 6 show the hit ratios of each base station. The LFU and LRU caching policies reached averages of 35% and 27%, respectively. The overlap-aware caching policy reached an average of 51%.

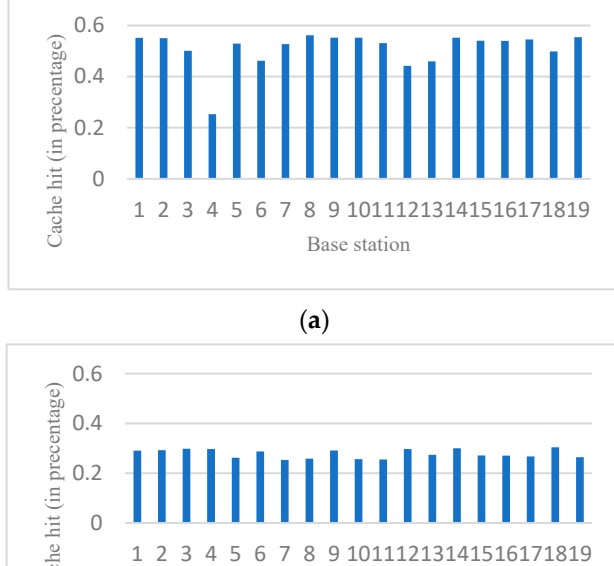

(a)

(b)

**Figure 6.** *Cont.*

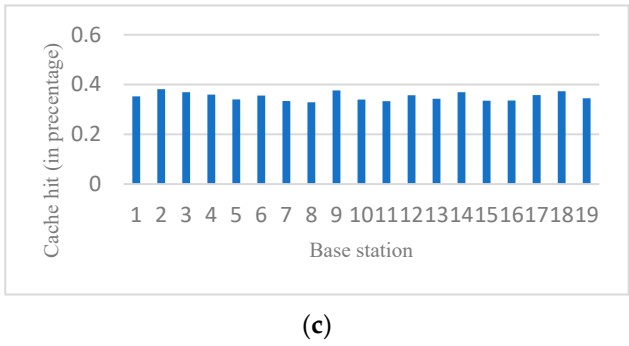

(**c**)

**Figure 6.** (**a**) Result of cache hit for OA. (**b**) Result of cache hit for LRU. (**c**) Result of cache hit for LFU.

The LFU and LRU policies remove those tiles that have not been requested for the longest time and those tiles with the fewest requests. With selection according to the mean and standard deviation of the yaw and pitch, tiles outside of the FoV can be selected for caching at the base station, which increases the hit ratio of the proposed policy.

*7.2. FoV Coverage*

In a multi-connection scenario, a missing cached tile can be transmitted from a nearby base station. FoV coverage is one of the scores in the objective function, and it represents the percentage of required tiles that are kept at a base station, out of all tiles. A higher percentage of coverage leads to a good viewing experience. We selected two users with different movements. One of the heads rotates more frequently. The frequency of the other head's rotation is gentler. The results in Figure 7 show that the Overlap-aware policy had 87% coverage for the user with slow movement and 54% coverage for the user with fast movement. The LRU and LFU caching policies had only 57% and 53% coverage, respectively, for the user with slow movement and only 26% and 27% coverage, respectively, for the user with fast movement.

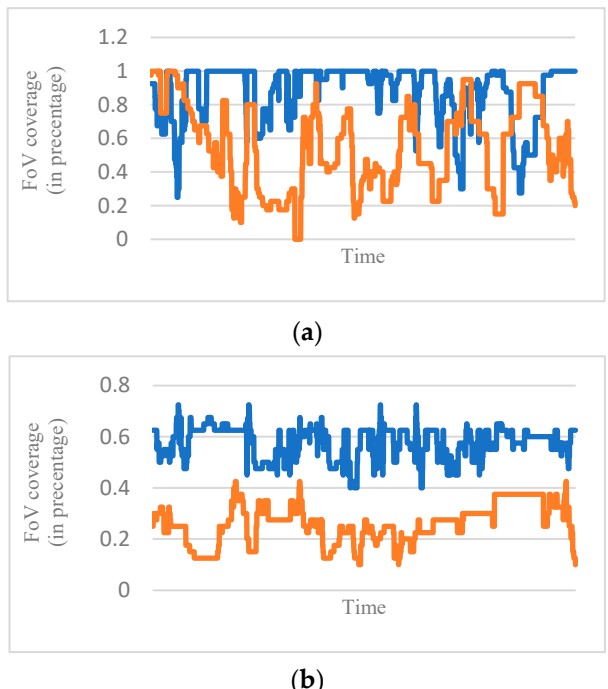

(**a**)

(**b**)

**Figure 7.** *Cont.*

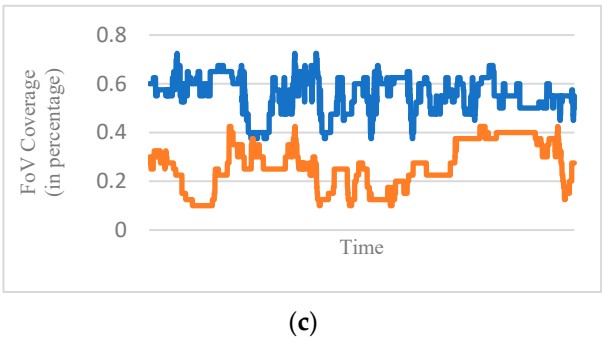

**(c)**

**Figure 7.** (**a**) Result of FoV coverage for OA. (**b**) Result of FoV coverage for LRU. (**c**) Result of FoV coverage for LFU. Blue line represents the user with gentle head rotation, and orange line represents the user with more frequent rotation.

In the Overlap-aware algorithm, we set the high-requirement tiles to high priority. In other words, the high-priority tiles represent compelling content. An example might be the stage at a concert.

*7.3. Caching Cost*

The caching cost is another score of the objective function. As the demand for video gradually increases, caching resources become more and more important. The results in Figure 8 show that the caching cost under the overlap-aware policy reached an average of 581. Under both the LRU and LFU caching policies, the caching cost reached an average of 781 in MEC.

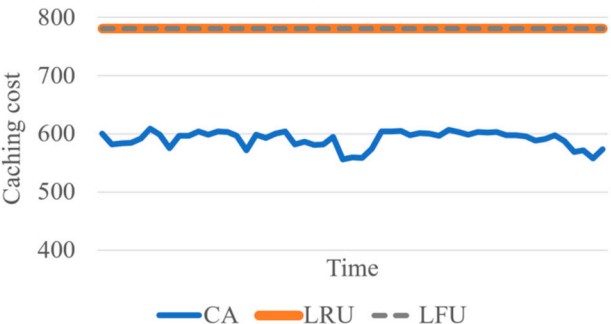

**Figure 8.** Result of caching cost.

## 8. Conclusions

Our research aims to create an interactive VR application for educational purposes on mobile network. To achieve this, we developed an algorithm that optimizes cache resource usage and user window coverage in a tile-based streaming architecture. Compared to the LRU and LFU algorithms, our approach reduces cache resource usage by 25% and expands window coverage by 30%, thus improving both cache resource and user window coverage. In this interactive VR application, teachers can deploy multiple scenarios, including 360° video. In tile-based 360 live streaming, caching takes on an important role. The correct caching policy can reduce not only the required storage but also the required bandwidth of the backhaul link. With multiple connected users, the tiles can be shared with users watching the same scene. In this paper, we proposed an overlap-aware caching strategy by deciding the priority of each tile and choosing the base station for caching based on this priority. Our policy results in a lower storage requirement and greater FoV coverage than LRU and LFU. Furthermore, our research also has designed a mobile edge computing (MEC) solution to enhance the performance of tile-based 360° video streaming by offloading some processing and storage tasks to the mobile edge. By offloading these

tasks to the mobile edge, the streaming server can deliver tiles more efficiently and with lower latency, resulting in a better overall user experience.

**Author Contributions:** Conceptualization, S.-M.C.; Methodology, S.-M.C. and E.H.-K.W.; Software, C.-S.C.; Validation, S.-M.C.; Investigation, E.H.-K.W.; Writing—original draft, C.-S.C.; Writing—review & editing, S.-M.C.; Supervision, E.H.-K.W. All authors have read and agreed to the published version of the manuscript.

**Funding:** This research received no external funding.

**Data Availability Statement:** The data is unavailable due to ethical restrictions.

**Conflicts of Interest:** The authors declare no conflict of interest.

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
