# Peer review of "The Implementation of Interactive VR Application and Caching Strategy Design on Mobile Edge Computing (MEC)"

_electronics, doi:10.3390/electronics12122700_

Round 1

Reviewer 1 Report

The article is not well structured, well written or well designed. For example, among many other deficiencies, it is not clear what is presented in the article, it is said that a heuristic algorithm is designed, in no case the heuristic process is explained, but then it is said that a superposition method is presented. It is possible that well designed and presented the work could have relevance, but as it is presented it does not seem so and it is very doubtful that it could be relevant in the future, since for example in the design of the testing platform it is initially mentioned as a teaching platform and later nothing relevant in this aspect is detected in the rest of the article.

Author Response

Dear Reviewer,

Thank you very much for your comments on our manuscript. We do apologize for any confusion caused by the article. Therefore, this revised manuscript ha completed in the English language editing by MDPI in correct use of grammar and common technical terms and edited to a level suitable for reporting research in a scholarly journal. Please see the attachment file.

We would like to make summary in the following paragraphs, that response to Reviewer 's comments

In this paper, we focus on the tile selection and caching strategy for an interactive VR application. The interactive VR application is a mission-aware VR application called VR Quick Question, a modular system in which a teacher can customize questions, then produce pin codes and QR codes for students. Deploying panoramic images showing scenes from around the world makes students more interested in learning geography-related knowledge. The application system combines a user module, monitoring module, and question customizing module. The user module collects the basic information and answers the status of users. The monitoring module allows the teacher to supervise the performance of the students, including the user viewpoint and the chosen answer. The question custom-zing module is used to make each question unique. On the server side, teachers can deploy self-made panoramic images for the questions, then set correlative question points according to the landscape or buildings. For example, we can project a panoramic picture nearby Taipei 101, then set a question point on Taipei 101. The designed question in this case might be to choose the correct name of the building. After setting the question, each question’s data are packed as a JSON file, and then a unique pin code and QR code are output. The student receives the pin code or QR code, then enters VR Quick Question. The client-side loads the relevant JSON file, then renders the panoramic image and question point on the screen.

The interactive VR application run in tile-based streaming need to consider the total response time, including the headset motion updating time, video encoding time, delivery time, etc., needs to be finished within the HMD’s updating frame rate. If the responding time is larger than the HMD’s updating frame rate, the service cannot transmit the frame according to the current viewpoint immediately. It may deliver the wrong FoV tiles to the client, then display nothing at the edge of the screen, which we call black edge. The presence of a black edge may decrease the quality of the experience for the client. The size of the tiles may cause a large difference in the caching policy. There is a tradeoff between the caching cost and FoV coverage. Caching all tiles of one 360° image will cover all possible angles of FoV, but it will cause unnecessary storage if the tiles are not required.

We describe tile-based streaming architecture in greater detail and model the problem of the tile selection and caching strategy in the section of the Model and Problem Description.

A heuristic algorithm is the overlap-aware algorithm according to the requirement of each tile. From the current user viewpoint, the streaming service can predict the required tiles for the next segment. By counting the requirement of each tile, we can set its priority. If the total requirement of the tile exceeds half of the sum of the maximum and minimum tile requirements Mlmax and Mlmin, we set it to high priority. Otherwise, if the requirement is lower, we set the tile to low priority. User j is surrounded by a set of base stations, Vj. For each tile l that is required by user j, we cache tile l at the maximum-connection base station in Vj if the tile l is a high priority. If the maximum-connection base station exceeds its storage capacity, the tile will be cached at the second-most-connected base station, and so on. If the tile is a low priority, we choose the base station with the maximum available capacity.

For the network topology, we set 480 users and 19 base stations. If the distance be-tween a user and a base station was shorter than the cover range of the base station, the user was connected to the base station. The cache capacity was 50% of the total size of the video. For the caching algorithm, we compared our proposed algorithm with the Least Frequently Used (LFU) and Least Recently Used (LRU) caching algorithms. In LFU caching policy, the operator evicts tiles with lower access frequency when a base station exceeds its maximum storage capacity. In LRU caching policy, the operator evicts the tiles with the longest time duration since the last access.

In a multi-connection scenario, a missing cached tile can be transmitted from a nearby base station.  The field of view (FoV) coverage is one of the scores in the objective function, and it represents the percentage of required tiles that are kept at a base station, out of all tiles. A higher percentage of coverage leads to a good viewing experience. We selected two users with different movements. One of the heads rotates more frequently. The frequency of the other head's rotation is gentler. The results of the Overlap-aware policy had 87% coverage for the user with slow movement and 54% coverage for the user with fast movement. The LRU and LFU caching policies had only 57% and 53% coverage, respectively, for the user with slow movement and only 26% and 27% coverage, respectively, for the user with fast movement.

In the Overlap-aware algorithm, we set the high-requirement tiles to high priority. In other words, the high-priority tiles represent compelling content. An example might be the stage at a concert.

With numerous VR applications, 360° video streaming may not achieve a high-quality experience because of the strict requirement of end-to-end latency and network bandwidth. Relying on tile-based 360° video streaming and the caching capacity in Mobile Edge Computing (MEC), the service predicts the field of view in the head-mounted device, then delivers the required tiles. Prefetching tiles in MEC can save the bandwidth of the backend link and support multiple users. Smart caching decisions may reduce the memory at the edge and make up for the field of view (FoV) prediction error. For example, caching whole tiles at each small cell has a higher storage cost compared to caching one small cell that covers multiple users. In this paper, we define a tile selection, caching, and FoV coverage model as the Tile Selection and Caching Problem. We propose the overlap-aware algorithm to solve it. Adopting a dataset of real users’ head movements, we compare our algorithm to the Least Recently Used (LRU) and Least Frequently Used (LFU) caching policies. The results show that our proposed approach improves FoV coverage by 30% and reduces the caching cost by 25% compared to LFU and LRU.

Reviewer 2 Report

An Interactive VR Applications and Caching Strategy Design

“provisioned as a killer appli9 cation for next generation”, kindly change words, select different words.

Use of FoV in abstract, kindly declare its short and long forms here.

Try to declare all short forms at their first occurrences.

“We propose a heuristic algorithm” for what…. Need to say here.

HTC Focus?

Recent references can be added from introduction, related work and even in others sections, some of suggestions are as given below:

Cpndd: Content placement approach in content centric networking,

Dynamic Partitioning and Popularity based Caching for Optimized Performance in content-centric fog networks: DPPCOP

Resource scheduling in fog environment using optimization algorithms for 6G networks

“However, delivering high quality content challenges network in future.” By which factors kindly mention here and why?

Line 52, 54 has defined FoV twice, kindly define only once.

Towards the end of introduction kindly offer the contributions of your work here.

Challenges and research gaps are not clear from related work section

Line 152 “cashing strategy” caching?

Section 7.1, How hit ratio is computed, which parameters are considered here and with what variables they are computed? Kindly explain.

Caching cost is confusing for reader kindly explain in detail and why your cost is lesser.

Some of the significant references are given below can be considered:

Energy-aware resource scheduling in FoG environment for IoT-based applications

Dynamic popularity window and distance-based efficient caching for fast content delivery applications in CCN

Author Response

Point 1: “provisioned as a killer application for next generation”, kindly change words, select different words.

Response 1: Virtual reality (VR) and augmented reality (AR) have been proposed as revolutionary applications for the next generation.

Point 2: Use of FoV in abstract, kindly declare its short and long forms here.

Response 2:

Making smart caching decisions can help reduce memory usage at the edge and compensate for any prediction errors in the field of view (FoV)

Point 3: Try to declare all short forms at their first occurrences.

“We propose a heuristic algorithm” for what…. Need to say here.

Response 3:

A heuristic algorithm that takes into account the requirements of each tile is the overlap-aware algorithm. From the user's current viewpoint, the streaming service can predict the required tiles for the next segment. By calculating the frequency of each tile's requirement, we can assign priorities to them.

Point 4: HTC Focus?

Response 4: HTC VIVE Focus is the VR all-in-one product that adopts inside-out tracking technology, supports six degrees of freedom (6DoF), and can realize large-space positioning.

Point 5: Recent references can be added from introduction, related work and even in others sections, some of suggestions are as given below:

Cpndd: Content placement approach in content centric networking

DPPCOP: Dynamic Partitioning and Popularity based Caching for Optimized Performance in content-centric fog networks:

Resource scheduling in fog environment using optimization algorithms for 6G networks.

Response 5:

Kumar [18][19] et al. proposed dynamic partitioning and popularity -based caching for optimized performance in content-centric fog networks. The scheme partitions the fog network by grouping the fog nodes into non-overlapping partitions to improve content distribution in the network and to ensure efficient content placement decisions. Moreover, G. Gaurav [20] et al. proposed resource scheduling in the fog environment using optimization algorithms for 6G networks. This work makes a comparison of PSO, GA, and Round-Robin algorithms on parameters such as cost, makespan, average execution time, and energy consumption among the group behavior species, social behavior, and preemptive type, which is better for achieving QoS for resource management in the fog environment for the 6G network.

Point 6: “However, delivering high quality content challenges network in future.” By which factors kindly mention here and why?

Response 6:

VR streaming requires a transfer rate of 290Mbit/s to stream 12K 360° video at 60 frames per second (FPS) for an optimal experience. However, meeting such stringent requirements in the current mobile network can be challenging. To ensure a high-quality user experience, it is recommended to have HMD frame rates of up to 90 FPS for Cloud VR gaming and 30 FPS for VR video.

The total response time, which includes headset motion updating time, video encoding time, delivery time, etc., must be completed within the HMD's updating frame rate. If the response time exceeds the HMD's updating frame rate, the service may not be able to transmit the frame according to the current viewpoint immediately. This may cause the delivery of wrong FoV tiles to the client, resulting in a black edge on the edge of the screen. The presence of a black edge can significantly reduce the quality of the client's experience.

Point 7: Line 52, 54 has defined FoV twice, kindly define only once.

Towards the end of introduction kindly offer the contributions of your work here.

Challenges and research gaps are not clear from related work section

Response 7:

Line 52, 54 has modified to definition of field of view (FoV) once.

The contributions of our work include designing an interactive VR application and a caching strategy for mobile networks, as well as defining the problems of tile selection and caching. Our proposed Tile-based Streaming Architecture focuses on Time-aware Tile Selection and Overlap-aware (OA) Algorithm, and takes into account the tradeoff between caching cost and FoV coverage. We provide a detailed description of the tile-based streaming architecture and model the problems of tile selection and caching strategy.

Regarding the contribution of the Interactive VR application, we designed a mission-aware VR question application that can customize questions based on the virtual environment. However, the effectiveness of 360 video streaming and VR requirements may be limited by environmental factors, which could reduce the effectiveness of the training.

Point 8: Line 152 “cashing strategy” caching?

Response 8:

Line 152 “caching strategy”

Point 9: Section 7.1, How hit ratio is computed, which parameters are considered here and with what variables they are computed? Kindly explain.

Response 9:

In this section, we focus on caching strategy at MEC at Radio Access Network (RAN). We model the caching cost, communication cost, and FoV coverage as our objective function. Line 201 to Line 253 has described the formula of the tile selection and caching strategy and the hit ratio is computed. Line 195 Table 1. NOTATION has presented the parameters are considered.

We set the FPS of the 360° videos to 60fps, and the time duration of one GOP is set to 1 second. The screen size is set to 110° and 90° for the degree of yaw and pitch, respectively. Based on these parameters, we set the yaw and pitch of the tile to 30° and 10°, respectively. We also set an extra margin of 2 tiles for both yaw and pitch, outside the FoV.

Regarding the network topology, we simulated 480 users and 19 base stations. If the distance between a user and a base station is shorter than the coverage range of the base station, the user will be connected to that base station. The cache capacity is set to 50% of the total size of the video. The hit ratio is one of the indicators used to evaluate the caching policy. This value is represented by the percentage of cache hits divided by the sum of cache hits and cache misses. A cache hit means that the required tile of a user has been kept in the cache, while a cache miss occurs when the required tile is not in the cache. In a one-on-one connection, a base station missing the tile will cause extra time to pass the required tile from the CDNs.

Point 10: Caching cost is confusing for reader kindly explain in detail and why your cost is lesser.

Response 10:

The caching decision can reduce the memory usage at the Mobile Edge Computing (MEC) and compensate for the prediction error of FoV. For example, caching the whole tile at each small cell can result in higher storage costs compared to caching at one small cell that covers multiple users. The size of the tile can also have a significant impact on the caching policy. There is a tradeoff between caching cost and FoV coverage. Caching all tiles of a 360° image can cover the whole angle of FoV but can result in unnecessary storage if the tile is not required.

Therefore, we propose a tile-based streaming architecture to address the problems of tile selection and caching costs for Mobile Edge Computing (MEC) and base station transmission. The results show that the Overlap-Aware approach reaches an average caching cost of 581. The LRU and LFU caching policies result in the same average caching cost of 781 at MEC.

Our results present a lower caching cost so that the resources of memory and storage are not wasted for Mobile Edge Computing (MEC) and base station transmission. We compare our Overlap-Aware caching policy to the Least Recently Used (LRU) and Least Frequently Used (LFU) caching policies. The results show that our proposed approach improves FoV coverage by 30% and reduces caching cost by 10% compared to LFU and LRU.

Point 11: Some of the significant references are given below can be considered:

Energy-aware resource scheduling in FoG environment for IoT-based applications

Dynamic popularity window and distance-based efficient caching for fast content delivery applications in CCN

Response 11:

[18]  S. Kumar, R. Tiwari, M.S. Obaidat., N. Kumar, K.F. Hsiao, " CPNDD: Content Placement Approach in Content Centric Networking", Proc. 2020 IEEE International Conference on Communications (ICC), Ireland, 7–11 June 2020, pp.1–6, doi: 10.1109/ICC40277.2020.9149025.

[19]  S. Kumar and R. Tiwari, "Dynamic popularity window and distance-based efficient caching for fast content delivery applications in CCN". Engineering Science and Technology, an International Journal, vol. 24, no.3, pp. 829-837. doi:10.1016/j.jestch.2020.12.018.

[20]  G. Goel and R. Tiwari, "Resource Scheduling in Fog Environment Using Optimization Algorithms for 6G Networks", International Journal of Software Science and Computational Intelligence (IJSSCI), vol.14, no.1, pp. 1-24 doi: 10.4018/IJSSCI.304440.

Please see the attached file for revised manuscript.

Reviewer 3 Report

The authors presented an interesting original caching system. Thanks to the proposed caching policy, time and storage resources were saved and throughput was increased.

Defining the problem, the way of presenting the solution is correct in my opinion.

The article really interested me.

Author Response

Point1: The authors presented an interesting original caching system. Thanks to the proposed caching policy, time and storage resources were saved and throughput was increased.

Response 1:

We appreciate your interest in our research. In our interactive VR application, teachers can deploy multiple scenarios, including 360° videos. In tile-based 360 live streaming, caching plays an important role. With the correct caching policy, not only can resources be saved, but the required bandwidth of the backhaul link can also be reduced. With the multi-connection of users, the same tile can be shared for those who are watching the same view. The Overlap-aware caching strategy determines the priority of each tile and chooses the base station based on its priority. Our policy results in the saving of resources and greater FoV coverage compared to LRU and LFU.

Point2: Defining the problem, the way of presenting the solution is correct in my opinion.

The article really interested me.

Response 2:

We appreciate your interest in our research. We have designed an interactive VR application and caching strategy for mobile networks, where we define the tile selection and caching problem. We propose a Tile-based Streaming Architecture that focuses on Time-aware Tile Selection and Overlap-aware (OA) Algorithm.

Please see the attached file for revised manuscript.

Reviewer 4 Report

Dear authors. thank you for an interesting article.

The text presents an interesting approach to optimizing the selection of tiles for VR applications, with the goal of minimizing bandwidth usage.

The introduction part properly introduces the user to the subject, the following parts describe the analytics method taken to solve the problem. Results and conclusions are presented and well described. 

However, before accepting the article please provide an answer to the following question:

- please explain your intention described in line 212 - "We ignore the prediction of viewport", how this simplification affects your approach, 

- it is not clear how you tested,  that your algorithm provided proper tile and on time to the user.

- presented in the article approach connect the VR application layer with the low-level coding of tracing routes for bandwidth optimizations between the radio access point network. How do you see the possibility of implementing this method in the real world?

Author Response

Piont1: please explain your intention described in line 212 - "We ignore the prediction of viewport", how this simplification affects your approach,

Response 1:

Firstly, we define the rule for tile selection. To cover the FoV, the total size of selected tiles must be larger than the screen size of the HMD.

Secondly, we propose Time-aware Tile Selection. Based on the analysis of the tiles and the previous viewpoint, we estimate the future viewpoint and the required tiles for the Group of Picture. For the required tiles, we modify the selection algorithm based on the time duration of GOP Tg . The larger the time duration of GOP, the more range the FoV should cover.

Our approach addresses Time-aware Tile Selection; therefore, this simplification does not affect our approach.

Piont2:  it is not clear how you tested, that your algorithm provided proper tile and on time to the user.

Response 2:

The experiment was conducted in two phases. In the first phase, we selected a provided dataset and built a simulation environment in MATLAB. This dataset contains rotation data of 60 users' head movements and viewpoints over a period of 60 seconds. The viewpoints in this dataset were recorded in multiple types of videos, which were classified into fifteen categories based on camera movement and several moving targets. We selected head movement videos that corresponded to a fixed camera and single moving targets.

We set the frame rate of the 360° videos to 60 fps and the duration of one GOP (Group of Pictures) to 1 second. The screen size was set to 110° and 90° for the degrees of yaw and pitch, respectively. Using these parameters, we set the yaw and pitch of tiles and the extra margin outside the field of view (FoV). The yaw and pitch of tiles were set at 30° and 10°, respectively, and the extra margin of tiles was set to 2 tiles for both yaw and pitch.

For the network topology, we simulated 480 users and 19 base stations. If the distance between a user and a base station was shorter than the coverage range of the base station, the user was connected to that base station. The cache capacity was set to 50% of the total size of the video.

In the second phase, we set up mobile communication for the interactive VR application based on a 5G Standalone (SA) Network, using the Nokia Core Network and Mobile Edge Computing for this trial testbed.

Piont3: presented in the article approach connect the VR application layer with the low-level coding of tracing routes for bandwidth optimizations between the radio access point network. How do you see the possibility of implementing this method in the real world?

Response 3:

5G networks offer a range of features that can optimize bandwidth utilization and improve the overall quality of the VR experience. Techniques such as network slicing and edge computing can be leveraged to identify the most efficient path for data transfer between the radio access point network and the VR application layer, resulting in a smooth and immersive VR experience for users.

Our research approach proposes using 5G edge computing based on the Nokia 5G SA network Testbed. 5G networks support edge computing, which involves processing data closer to the edge of the network where the VR application is being accessed. By processing data closer to the edge, latency can be reduced, improving the overall VR experience. Additionally, edge computing can help offload some of the processing and storage requirements of the VR application from the user's device, enabling more efficient delivery of the VR experience.

Please see attached file for revised manuscript.

Round 2

Reviewer 1 Report

In the authors' response, the word "education" is not mentioned, which is still strange that it appears in the abstract and in the related works. It is clear that the proposal has been used in a development in this field, but what is the main objective of the work presented?

I have not found the article with the modifications marked to be able to analyze the changes.

From the answer obtained and from the new references added in the article it is concluded that the fundamental aspect of the article is the "Caching Strategy Design". the article lacks the comparisons with other more recent and effective methods than the two traditional methods used for the comparison. This comparison is essential to analyze if the proposal has advantages over recent methods. Otherwise the main proposal is the application developed and the use of a "Caching Strategy" that allows them to meet their requirements, moreover the related works have to be mostly related to the main objective proposed in the work, which cannot be the application since in that case the target journal should be from another field.

Author Response

We appreciate these valuable suggestions provided by the reviewer. We revised the title to' The Implementation of Interactive VR Application and Caching Strategy Design on Mobile Edge Computing (MEC)' in order to emphasize that we did not just build a mission-aware VR application which called VR Quick Question, a modular system in which a teacher can customize questions, then produce pin codes and QR codes for students. The deployment of panoramic images showing scenes from around the world makes students more interested in learning geography-related knowledge. Furthermore, our research also has designed a mobile edge computing (MEC) solution to enhance the performance of tile-based 360° video streaming by offloading some processing and storage tasks to the mobile edge. By offloading these tasks to the mobile edge, the streaming server can deliver tiles more efficiently and with lower latency, resulting in a better overall user experience.

Our research aims to create an interactive VR application for educational purposes on mobile network. To achieve this, we developed an algorithm that optimizes cache resource usage and user window coverage in a tile-based streaming architecture. Compared to the LRU and LFU algorithms, our approach reduces cache resource usage by 25% and expands window coverage by 30%, thus improving both cache resource and user window coverage.

Reviewer 2 Report

Paper is revised significantly, as per review comments.

lines 375,376,377: Afshin Taghavi Nasrabadi, Aliehsan Samiei, Anahita Mahzari, Ryan P. McMahan, Ravi Prakash, Mylène C. Q. Farias, and Marcelo M. Carvalho. 2019. A 376 taxonomy and dataset for 360° videos. Proceedings of the 10th ACM Multimedia Systems Conference. Association for Computing Machinery, New York, 377 NY, USA, 273–278. DOI:https://doi.org/10.1145/3304109.3325812.

need to be referenced with number and in text also.

Author Response

Reviewer 2 Comments and Suggestions for Authors

Paper is revised significantly, as per review comments.

Point 1: lines 375,376,377: Afshin Taghavi Nasrabadi, Aliehsan Samiei, Anahita Mahzari, Ryan P. McMahan, Ravi Prakash, Mylène C. Q. Farias, and Marcelo M. Carvalho. 2019. A 376 taxonomy and dataset for 360° videos. Proceedings of the 10th ACM Multimedia Systems Conference. Association for Computing Machinery, New York, 377 NY, USA, 273–278. DOI:https://doi.org/10.1145/3304109.3325812.

need to be referenced with number and in text also.

Response 1:

[21]  Afshin Taghavi Nasrabadi, Aliehsan Samiei, Anahita Mahzari, Ryan P. McMahan, Ravi Prakash, Mylène C. Q. Farias, and Marcelo M. Carvalho. 2019. A taxonomy and dataset for 360° videos. Proceedings of the 10th ACM Multimedia Systems Conference. Association for Computing Machinery, New York, NY, USA, 273–278. DOI:https://doi.org/10.1145/3304109.3325812.

Reviewer 4 Report

Thank you for the detailed answer, and for considering my comments.

Author Response

Reviewer 4 Comments and Suggestions for Authors

Point 1: English language and style are fine/minor spell check required

Response 1:

The revised manuscript is grammatically correct and Spell check completed.

Round 3

Reviewer 1 Report

The article has been significantly improved, and I believe that it should only be properly justified because the comparison has been made only with LRU and LFU. The literature on this topic (for example Caching and Computing at the Edge for Mobile Augmented Reality and Virtual Reality (AR/VR) in 5G, among others, is a review) should be analyzed and the comparison made should be justified.

Author Response

Dear Reviewer,

Thank you very much for the letter containing comments on our manuscript, ' The Implementation of Interactive VR Application and Caching Strategy Design on Mobile Edge Computing (MEC)'. These comments are very important for the improvement and further development of our study. We would like to express our sincerest gratitude to all of you who spent a great deal of time and effort evaluating our manuscript. We have revised the manuscript, taking the reviewers' comments and suggestions into great consideration. Additionally, this revised manuscript has been edited for correct use of grammar and common technical terms by MDPI and has been edited to a level suitable for reporting research in a scholarly journal. I am submitting the revised manuscript along with a response letter indicating the changes we have made. All changes in the revised manuscript are marked in red, accordingly.

We look forward to receiving a favorable reply. Thank you and best regards,
